# CSATTENTION: CENTROID-SCORING ATTENTION FOR ACCELERATING LLM INFERENCE

## ABSTRACT

Long-context LLMs increasingly rely on extended, reusable prefill prompts for agents and domain Q&A, pushing attention and KV-cache to become the dominant decode-time bottlenecks. While sparse attention methods reduce computation and transfer costs, they struggle to simultaneously maintain model accuracy at high sparsity levels due to the inherent distribution shift between Queries and Keys. To address this challenge, we propose Centroid-Scoring Attention (CSAttention), a training-free sparse attention method optimized for high-throughput serving of reusable contexts. CSAttention adopts a *storage-for-computation* strategy: it leverages query distributions to construct a query-centric lookup table in each subspace during the offline prefill stage, enabling online decoding to perform efficient searches and centroid-score accumulation that amortize the one-time indexing cost. By combining subspace partitioning with query-centric table construction, CSAttention mitigates distribution shift between queries and keys, and reliably recovers high-scoring keys even under very high sparsity, enabling significant computational savings while maintaining competitive model performance. Extensive experiments demonstrate that CSAttention maintains *near-lossless* model accuracy while delivering substantial improvements in inference efficiency. Compared to state-of-the-art sparse attention methods, CSAttention achieves superior model accuracy and higher inference speed in high-sparsity (95%) and long-context (32K-128K) scenarios. Notably, CSAttention achieves up to $4.24\times$ speedup over full attention when decoding 128K context length, demonstrating its practical value for scalable cache-enabled long-context inference.

## 1 INTRODUCTION

Long-context LLM usage is accelerating, driven by LLM agents and domain Q&A workflows that demand *very long prompts*. In many of these deployments, workloads naturally split into an *offline prefill* stage and an *online decode* stage (Lu et al., 2024; Jin et al., 2024; Gao et al., 2024; Lee et al., 2025). This separation is particularly evident in "Write-Once, Read-Many" scenarios (e.g., Context Caching for RAG or Agent schemas), where servers run a one-time prefill to materialize reusable KV and prepare search auxiliaries, persisting them outside HBM (CPU DRAM or SSD) for later reuse. Online decode is the request-time path: when a user query arrives, the system loads the needed artifacts on demand and performs decoding over the growing context; lightweight maintenance (e.g., appending new keys) is allowed as long as latency remains predictable. With this separation in mind, the prefill step may feed tens of thousands to millions of tokens, amortizing its cost over thousands of subsequent decoding steps that repeatedly apply attention over the accumulated context.

Beyond the quadratic cost of dense attention, a second bottleneck dominates in practice: the KV cache. Its footprint scales linearly with sequence length, layers, and heads; storing all keys/values on HBM quickly becomes the limiting factor for throughput, often forcing systems to page KV to CPU RAM and back during decoding. Production serving stacks (e.g., vLLM with PagedAttention) mitigate fragmentation and enable sharing but still incur bandwidth/latency costs as contexts grow (Kwon et al., 2023).

A concrete calculation underscores the challenge. The memory footprint of the KV cache grows linearly with the context length and model dimension, creating immense pressure on GPU memory. Even with grouped-query attention (GQA/MQA) reducing the count of KV heads (Shazeer, 2019;

Ainslie et al., 2023), long contexts remain daunting: for Llama-3-8B at 1M tokens, recent measurements report on the order of $10^2$ GB of KV memory in bf16 without approximation, routinely exceeding a single GPU (Luo et al., 2025).

A natural response is sparse attention: standard attention matrices exhibit inherent sparsity, wherein a large fraction of the computed weights are close to zero and can be pruned without significant impact on output quality (Zhang et al., 2025b). Therefore, the model can reliably attend to only a small fraction of keys, reducing attention FLOPs and the effective KV touched per step simultaneously. Prior work explores three main directions. (i) Token eviction/retention: dynamically keep only "heavy-hitter" tokens in the cache (e.g., $H_2O$ and follow-ups) (Zhang et al., 2023; 2024), which prunes storage but can be sensitive to online prediction errors. (ii) Bandwidth-aware fetching: techniques like SparQ selectively fetch historical KV to raise memory-bandwidth efficiency during attention (Ribar et al., 2024). (iii) Index-based retrieval: treat KV search as MIPS/ANN over quantized representations (e.g., PQCache) or use sampling via LSH (e.g., MagicPIG) to approximate attention (Zhang et al., 2025a; Chen et al., 2024). While these methods reduce computational and transfer costs, they encounter a fundamental *challenge at high sparsity*: existing indexes often rely on Key-based clustering, which suffers from Query-Key distribution shift, making it exceedingly difficult to simultaneously maintain high model performance and achieve fast inference speed.

In this work, we propose Centroid-Scoring Attention (CSAttention), a training-free sparse attention method that accelerates long-context serving. To achieve high model performance and fast inference under high sparsity, CSAttention adopts a *storage-for-computation* strategy: it leverages query distributions to construct a query-centric lookup table in each subspace during the offline prefill stage, enabling online decoding to perform efficient searches and centroid-score accumulation over regular, GPU-friendly data structures. Specifically,

- **Query-centric tables (offline).** Split the feature space into $m$ subspaces. For each subspace, cluster queries from prefill into $C$ centroids. For every centroid, precompute partial dot-products with all keys in that subspace as centroid-scores, and store a compressed Top-$L$ list (indices + scores). This design amortizes the one-time indexing cost across many requests that share the same long prefill context.

- **Keys retrieval (online).** For a new query, select its nearest centroid in each subspace (1-of-$C$ per subspace), fetch the $m$ short lists, and sum partial scores by key index on GPU. Keys truly aligned with the query tend to exhibit high centroid-scores across multiple subspaces, rising to the top after sparse accumulation—mitigating potential query drift via subspace aggregation without scanning the whole cache.

- **Middle-dominant scheduling.** We prioritize the middle region of the context (where recency heuristics are weakest) while merging a small recent window as passthrough to preserve short-range dependencies.

- **Streaming-friendly updates.** When a new key arrives, we try-insert it into each centroid's Top-$L$ if its partial score exceeds the current minimum, keeping maintenance overhead negligible and decode latency predictable.

**Why this helps.** (i) By exploiting query-centric clustering offline, the index structure tracks the geometry of queries $Q$ rather than only keys $K$, mitigating $Q/K$ distribution shift and ensuring robustness against query drift during generation. (ii) By utilizing compressed lookup tables and only running a small number of regular GPU kernels during decoding, CSAttention avoids per-query score movement and irregular control flow, sustaining high hardware utilization and inference speed. (iii) By employing subspace partitioning and query-centric tables, CSAttention effectively recovers high-scoring keys under very high sparsity (e.g., 95%), enabling significant computational savings while maintaining model accuracy.

**Results at a glance.** (i) *Near-lossless accuracy at 95% sparsity*: on LongBench evaluations across three models (Llama-3-8B, Qwen-8B, and Mistral-7B), CSAttention maintains nearly identical accuracy to full attention (within 0.7% loss) at 95% sparsity. (ii) *Best accuracy and speed over competitors*: under high sparsity (95%) and long-context settings (32K–128K), CSAttention outperforms state-of-the-art sparse attention methods in both model accuracy and inference throughput. (iii) *Scalable speedup over full attention*: the performance advantage of CSAttention increases with context length, reaching up to 4.24× speedup compared to standard FlashAttention-based baselines

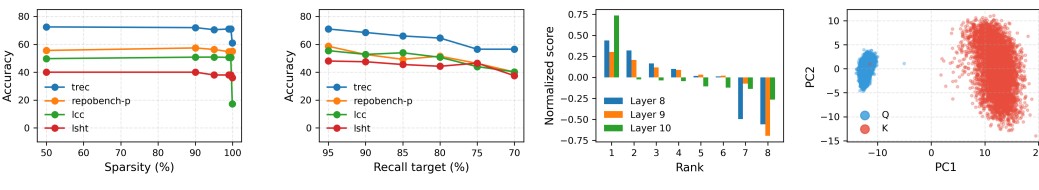

(a) Accuracy vs. sparsity  (b) Accuracy vs. recall  (c) Subspace rank–share (L8–10)  (d) Q/K PCA (L10, H0)

Figure 1: **Observations.** (a/b) Accuracy vs. sparsity/recall on four LongBench tasks (Llama-3.1-8B-Instruct) (c) Heterogeneous subspace: rank-share of accumulated $qk$ contributions across $m = 8$ subspaces, grouped over Layers 8–10. (d) PCA of queries $Q$ and keys $K$ from Llama-3.1-8B-Instruct (Layer 10, Head 0) shows a distribution shift between $Q$ and $K$.

at 128K context length. (iv) *Robustness in Long-Generation*: On LongBench-v2 with Chain-of-Thought (CoT) decoding, CSAttention preserves accuracy even as the generation length extends, validating the stability of query-centric clusters against distribution drift.

## 2 OBSERVATIONS AND MOTIVATION

### 2.1 PRELIMINARIES AND EMPIRICAL SPARSITY OF ATTENTION

Let $q, K, V \in \mathbb{R}^{N \times d}$ be the query, key, and value sequences for one head, with $d$ the hidden size and $N$ the (growing) context length during decoding. Scaled dot-product attention computes per-query weights

$$a = \text{softmax}\left(\frac{qK^\top}{\sqrt{d}}\right) \in \mathbb{R}^{1 \times N},$$

and returns the weighted value aggregation $o = aV \in \mathbb{R}^{1 \times d}$. The vector $a$ (the *attention scores*) sums to 1 and determines which past tokens' values contribute to the output. In multi-head attention (MHA/GQA/MQA), this is applied per head and concatenated or averaged across heads.

Attention matrices in long-context LLMs are effectively sparse (most scores are near zero)—so keeping only the Top-K keys typically preserves task quality. Let the keep ratio $\rho = \text{K}/N$ denote the fraction of keys scored/used for a query (sparsity $= 1 - \rho$). In Figure 1(a), we evaluate four LongBench subtasks on Llama-3.1-8B-Instruct, forcing attention to select only the Top-K keys while varying sparsity from 50% to 99.9% (i.e., $\rho$ from 0.5 down to 0.001). Accuracy remains essentially unchanged even at 95% sparsity. It indicates that retaining only the high-weight keys can substantially reduce attention computation without degrading quality, consistent with recent reports (Liu et al., 2024; Zhang et al., 2025b).

### 2.2 TOP-K RECALL GOVERNS ACCURACY IN HIGH SPARSITY

Even modest misses in the true Top-K hurt accuracy. Since the true Top-K is unavailable at decode time without full attention, any sparse-attention method can only *approximate* it. As shown in Figure 1(b), we first use full attention to obtain the oracle Top-K keys for each query, fixing $\text{K} = \rho N$ with $\rho = 0.05$, then enforce a target recall $r$ by randomly replacing the $(1-r)N$ keys with the next-best keys ($K+1$, $K+2$, ...). Accuracy degrades as recall drops—and the loss is already visible when recall falls from 95% to 90%. Thus, achieving consistently high Top-K recall is essential for sparse attention to match full-attention accuracy.

### 2.3 IMPORTANCE DEVIATION OF DIFFERENT SUBSPACES

Search-based sparse attention methods essentially transform the problem of attention selection into a vector similarity search task (Liu et al., 2024). Partitioning the $d$-dimensional origin space into $m$ $d/m$-dimensional subspaces and leveraging clustering to construct an index in each subspace is a highly effective scheme in the field of vector similarity search (Jegou et al., 2010; Wei et al., 2025). The underlying principle is that from the perspective of the original space, clustering with $l$ clusters

in $m$ subspaces is equivalent to the *Cartesian product* representation: $C_{total} = C_1 \times C_2 \times \cdots \times C_m$, where $C_i$ and $C_{total}$ denote the centroids set in each subspace and in the origin space. The clustering complexity is thus reduced from $O(l^m)$ to $O(ml)$, thereby improving the method performance.

The recently proposed *subspace collision* vector search framework has demonstrated promising performance (Wei et al., 2025). This framework assigns equal weight to each subspace within Euclidean space. However, we observe that subspaces contribute very unevenly to the inner-product attention score. We run LLAMA-3.1-8B-INSTRUCT on a random sample from the LONGBENCH subtask MULTIFIELDQA_ZH, collect queries $Q$ and keys $K$ after prefilling, and analyze layers 8–10. We split $Q$ and $K$ into $m$ equal $d/m$-dimensional sub-vectors, $Q = [Q^{(1)}; \ldots; Q^{(m)}]$ and $K = [K^{(1)}; \ldots; K^{(m)}]$. For each subspace $b$, we compute the contribution matrix $S^{(b)} = Q^{(b)}(K^{(b)})^\top$ and then mean-normalize across subspaces to obtain per-subspace shares. As shown in Figure 1(c), the subspace shares are highly skewed, indicating that the different importance of each subspace.

It is necessary to design customized search strategies for attention calculation, so as to meet the following requirements: (i) achieving high efficiency by partitioning the subspace, and (ii) achieving high precision by considering the importance deviation of different subspaces.

**Implication.** Attention-friendly search should preserve the efficiency benefits of subspace partitioning *and* account for unequal subspace contributions—e.g., via subspace-aware scoring, weighting, or prioritized probing—to maintain high recall at extreme sparsity.

### 2.4 SEARCH PATH: FROM KEY-CENTRIC TO QUERY-CENTRIC

Prior search-based sparse attention methods only use keys to build clustering-based indices during the prefilling stage (Zhang et al., 2025a; Liu et al., 2025). During decoding, these methods typically follow a *key-centric* search path: $Q \rightarrow K$-centroid$\rightarrow K$. However, we run Llama-3.1-8B-Instruct on a random NarrativeQA example and visualize the layer-10, head-0 query and key vectors via PCA, as shown in Figure 1(d). The distributions of $Q$ and $K$ diverge significantly. This misalignment stems from the fact that $Q$ and $K$ are generated by different projections, which can be *biased* for certain heads/timesteps, especially under stylistic/domain shifts. Key-only indices (built on $K$) can become out-of-distribution (OOD) for $Q$ to search, causing unstable recall at high sparsity.

An *query-centric* search path, denoted as $Q \rightarrow Q$-centroid$\rightarrow K$, offers greater stability. Since the nearest-centroid assignment occurs *in the same space* as $Q$, it significantly reduces OOD risk. Once the query-centric centroid is selected, the search strategy only accesses the precomputed $K$ lists associated with that centroid-eliminating the need for an additional $Q \rightarrow K$ centroid hop during decoding-thereby notably improving recall stability under high sparsity.

## 3 METHODOLOGY

### 3.1 OVERVIEW OF CSATTENTION

**Architecture overview.** Figure 2 provides an overview of CSAttention, which consists of an offline prefilling stage and an online decoding stage. To enhance inference efficiency, CSAttention employs a subspace partitioning strategy, as analyzed in Section 2.3. We split $d$ dimensions into $m$ subspaces with sizes $\{d_b\}_{b=1}^m$ and $\sum_b d_b = d$. For key $k_i \in K = [k_1, k_2, ..., k_N]$ and query $q$,

$$k_i = \left( k_i^{(1)}, \ldots, k_i^{(m)} \right), \quad q = \left( q^{(1)}, \ldots, q^{(m)} \right), \quad qk_i^\top = \sum_{b=1}^m \underbrace{q^{(b)}(k_i^{(b)})^\top}_{\text{subspace partial}}.$$

During prefilling, operations are only performed independently within each subspace: queries are clustered, and the inner products between each centroid and all keys are computed and recorded as *centroid-scores* in a lookup table. During decoding, a query-centric search is conducted inside each subspace, after which centroid-scores are sparsely accumulated across subspaces to efficiently retrieve the most critical tokens.

**Design overview.** (i) Subspace split. We use uniform split by default ($d_b = d/m$) for balanced GEMV/GEMM sizes; nonuniform splits are possible when heads emphasize bands. (ii) Normalization. We $\ell_2$-normalize subspace vectors when clustering and during centroid matching (cosine

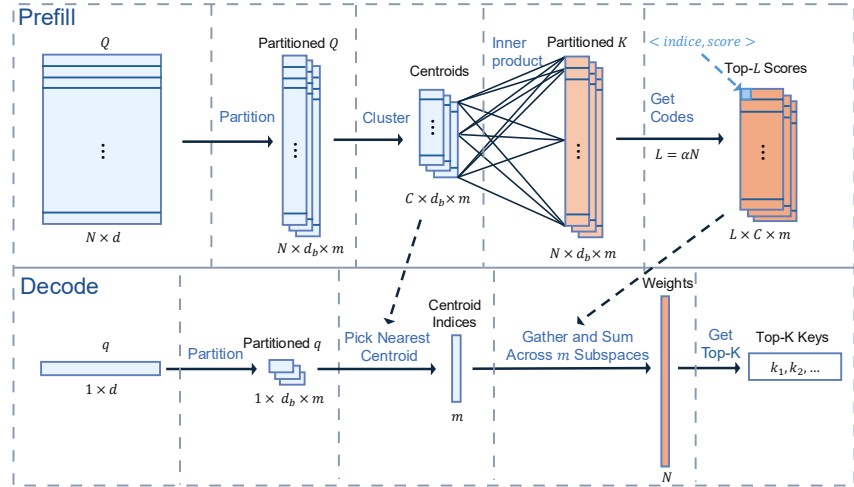

Figure 2: **CSAttention overview.** Prefill (top): partition queries, cluster per subspace on GPU, score each query-centroid against all keys in that subspace, and store fixed Top-$L$ (idx, score) lists. Decode (bottom): partition the new query, pick the nearest centroid per subspace, fetch $m$ lists, gather-and-sum across subspaces on GPU (or search on CPU in the CPU↔GPU variant), then take Top-K and run standard KV gather/attention.

scores); the model's native scaling is preserved for attention. (iii) Fixed-size tables. Each centroid $(b, j)$ stores contiguous arrays $\mathcal{I}_j^{(b)} \in \mathbb{N}^L$ and $\mathcal{V}_j^{(b)} \in \mathbb{R}^L$ for indices (as int32) and scores (as fp16); this guarantees coalesced loads and a bounded decode union ($\leq mL$). (iv) Middle-dominant schedule. Search is applied to the middle region; a recent window of size $R$ is merged as passthrough to preserve short-range dependencies.

### 3.2 OFFLINE (PREFILL): QUERY CLUSTERING AND PER-CENTROID SCORING

**(1) Subspace partition and queries clustering.** Given prefill queries $Q \in \mathbb{R}^{N \times d}$, partition each $q$ into $\{q^{(b)}\}_{b=1}^m$ and run mini-batch $k$-means *on GPU* per subspace to obtain $C$ centroids $\{c_j^{(b)}\}_{j=1}^C$:

$$\min_{\{c_j^{(b)}\}} \sum_{q \in Q} \min_{j \in [1..C]} \left\| \frac{q^{(b)}}{\|q^{(b)}\|_2} - c_j^{(b)} \right\|_2^2, \quad \text{s.t. } \|c_j^{(b)}\|_2 = 1.$$

We use cosine $k$-means (normalize vectors); seeds are $k$-means++ with a small number of iterations (e.g., 10–20) sufficient for stable nearest assignments at decode.

**(2) Per-centroid scoring in each subspace.** For each centroid $(b, j)$ we compute

$$s_j^{(b)}(i) = c_j^{(b)}(k_i^{(b)})^\top, \qquad i \in [1..N],$$

via batched GEMM across centroids, where $N$ denotes the sequence length; then keep Top-$L$ pairs $(i, s_j^{(b)}(i))$ and serialize into $(\mathcal{I}_j^{(b)}, \mathcal{V}_j^{(b)})$ on the target device (GPU for All-GPU; CPU DRAM for CPU↔GPU). Tables are *fixed size* and reused across requests that share the long prefill.

### 3.3 ONLINE (DECODE): QUERY-CENTRIC SEARCH AND SPARSE ACCUMULATION

Given a new query $q$:

**(1) Nearest query-centroid per subspace.** For each subspace $b$,

$$\hat{j}_b = \arg \max_{j \in [1..C]} \tilde{q}^{(b)}(c_j^{(b)})^\top, \quad \tilde{q}^{(b)} = q^{(b)} / \|q^{(b)}\|_2.$$

Implementation is a batched GEMV over $C$ centroids per subspace (per head), mapping well to GPU.

**(2) Gather $m$ short lists and build the union.** Fetch $\{(\mathcal{I}_{\hat{j}_b}^{(b)}, \mathcal{V}_{\hat{j}_b}^{(b)})\}_{b=1}^m$. Concatenate into a single array of at most $mL$ pairs; loads are contiguous/coalesced.

**(3) Reduce-by-key (gather-and-sum).** Radix-sort the concatenated pairs by index and perform a segmented sum to compute

$$\hat{s}(i) = \sum_{b=1}^m w_b \Big[ i \in \mathcal{I}_{\hat{j}_b}^{(b)} \Big] \cdot \mathcal{V}_{\hat{j}_b}^{(b)}[i], \qquad i \in U,$$

with $|U| \leq mL$. We use uniform subspace weights ($w_b = 1$), learned or confidence-based $w_b$ are possible but not required in our best settings. This step is branchless and implemented with warp-synchronous reductions. This step realizes *centroid-scoring*: aggregated scores are sums of precomputed centroid→key partials, avoiding any online $Q{\to}K$ code movement.

**(4) Merge recent window and select Top-K.** Union the recent window $\{N - R + 1, \ldots, N\}$ and select Top-K on device. Only these K keys are used in attention; others are ignored.

**(5) Streaming updates.** After attention, when a new key $k_S$ is appended, compute score$_{b,j} = (c_j^{(b)})^\top k_S^{(b)}$ for all $(b, j)$ and try-insert into $(\mathcal{I}_j^{(b)}, \mathcal{V}_j^{(b)})$ if it exceeds the current minimum. In CPU↔GPU mode, this maintenance runs on CPU asynchronously while GPU executes attention.

### 3.4 Execution Modes and Memory/Cost Considerations

**CPU↔GPU (index and KV in DRAM; asynchronous execution).** When HBM capacity is constraining, both the KV cache and the centroid tables are resident in CPU DRAM, and decoding proceeds with an *explicitly overlapped* CPU–GPU pipeline. *Prefill:* on the device, Q/K/V projections and full prefill attention execute on the default stream, while per-subspace query clustering and centroid→key partial scoring run on a dedicated compute stream; the resulting tensors are transferred to host *via non-blocking D2H* into pinned buffers, and keys/values are appended to a host-resident KV store. Streams are synchronized only once before the first decode step. *Decode (per step):* (1) the CPU performs the bounded search by merging the $m$ Top-$L$ lists, accumulating centroid scores by index, and selecting Top-K ( K=$\rho S$, typically $\rho{\approx}0.05$ ); (2) the corresponding K keys/values are gathered from the host KV store and *asynchronously* copied H2D; (3) the GPU runs attention on the selected set. Streaming updates insert the newly appended key into each centroid's Top-$L$ on the CPU without resizing tables. *Overlap:* GPU attention at step $t$ overlaps with CPU search and the H2D transfer for step $t{+}1$ using separate CUDA streams and events, so PCIe latency is largely hidden. Because only $O(K)$ vectors are moved per step, the transfer budget is deterministic and small.

**All-GPU (index and KV on HBM).** When HBM is sufficient, we keep both tables and KV on-device. Decode is: nearest-centroid (batched GEMV) $\rightarrow$ coalesced list fetch $\rightarrow$ reduce-by-key (radix sort + segmented sum) $\rightarrow$ device Top-K $\rightarrow$ attention over the gathered K pairs. No per-query score movement; kernels are regular and easily batched, so speedups appear already at moderate contexts. All kernels are regular (batched GEMV, contiguous list loads, radix sort, segmented reduction, device-side Top-K), which preserves high occupancy and minimizes control-flow divergence.

## 4 Experiments

### 4.1 Setting

**Models & baselines.** We evaluate on three instruction-tuned backbones: Llama3-8B, Qwen3-8B, and Mistral-7B (Instruct v0.3). Baselines include MagicPIG (LSH sampling; $L{=}300$, $K{=}10$), SparQ Attention (bandwidth-aware fetching), H$_2$O (heavy-hitter retention), and PQCache (PQ-based KV retrieval; we give it 15 $k$-means iterations and SUBBITS$=8$ to favor accuracy at high sparsity). Unless stated otherwise, all methods target a comparable keep ratio near $5\%$.

| LongBench Evaluation Tasks | | | | | | | | | | | | | |
|---|---|---|---|---|---|---|---|---|---|---|---|---|---|
| **MQA-E** | **MQA-Z** | **NarQA** | **M-News** | **Musiq** | **Trec** | **Samsum** | **TrivQA** | **P-Ret** | **Hotpot** | **G-Rep** | **LCC** | **LSHT** | **VCSum** | **Avg** |
| **Llama-3.1-8B-Instruct** | | | | | | | | | | | | | |
| Full | 55.54 | 62.87 | 29.91 | 27.16 | 30.89 | 72.50 | 43.75 | 91.65 | 100.0 | 56.16 | 35.26 | 64.89 | 46.00 | 17.16 | 52.41 |
| CSAttention | **56.02** | 62.01 | 30.46 | 26.38 | **31.11** | 71.50 | **44.16** | **91.95** | 99.50 | 55.94 | 33.60 | 63.33 | 45.00 | 17.63 | 52.04 |
| PQCache | 52.96 | 57.57 | 30.14 | 16.67 | 28.69 | 71.00 | 40.01 | 91.82 | 99.00 | 55.22 | **34.01** | 60.52 | 43.00 | 16.38 | 49.79 |
| H2O | 40.17 | 40.01 | 29.21 | 23.94 | 28.08 | 62.00 | 41.10 | 90.32 | 97.00 | 53.36 | 28.32 | 57.96 | 23.50 | 16.50 | 45.11 |
| SparQ | 39.56 | 34.32 | 26.96 | 21.78 | 28.48 | 47.00 | 42.11 | 89.26 | 87.00 | 51.83 | 25.21 | 55.42 | 21.00 | 15.18 | 41.79 |
| MagicPig | 48.78 | 53.16 | 25.86 | 14.50 | 19.20 | 70.00 | 42.00 | 65.05 | 96.00 | 38.50 | 23.39 | 61.11 | 38.00 | 7.83 | 43.10 |
| **Qwen-8B** | | | | | | | | | | | | | |
| Full | 53.67 | 63.37 | 26.05 | 24.88 | 36.18 | 71.50 | 44.30 | 88.54 | 100.0 | 59.40 | 33.35 | 69.13 | 47.50 | 14.31 | 52.30 |
| CSAttention | 53.21 | **63.74** | 26.18 | **24.60** | 37.05 | 72.00 | 45.10 | 88.62 | 100.0 | 59.18 | 32.63 | 68.89 | 46.00 | 14.29 | 52.25 |
| PQCache | 51.98 | 60.35 | **26.98** | 21.90 | 36.90 | 72.00 | 42.80 | 84.10 | 100.0 | 58.73 | **33.09** | 61.01 | 44.00 | **14.39** | 50.59 |
| H2O | 50.10 | 57.90 | 26.01 | 23.99 | 34.20 | 61.00 | 44.50 | 85.00 | 98.50 | 54.12 | 29.99 | 63.01 | 31.00 | 13.21 | 48.04 |
| SparQ | 45.32 | 50.07 | 25.88 | 21.03 | 31.90 | 59.00 | 43.90 | 80.20 | 91.00 | 47.93 | 27.62 | 48.32 | 25.50 | 13.48 | 43.65 |
| MagicPig | 52.11 | 57.32 | 26.31 | 18.94 | 28.08 | 58.00 | 44.22 | 87.90 | 98.50 | 51.88 | 24.32 | 55.67 | 40.00 | 9.12 | 46.60 |
| **Mistral-7B-Instruct-v0.3** | | | | | | | | | | | | | |
| Full | 50.21 | 53.19 | 27.74 | 26.57 | 26.50 | 70.00 | 46.30 | 89.04 | 97.00 | 51.08 | 34.22 | 64.32 | 47.00 | 15.68 | 49.92 |
| CSAttention | 49.92 | **52.94** | 25.56 | **27.06** | 26.10 | 70.50 | **45.91** | 90.59 | 97.00 | 49.34 | 32.88 | 63.98 | 46.00 | 16.44 | 49.92 |
| PQCache | 45.57 | 39.59 | 22.57 | 26.04 | 22.30 | 71.00 | 42.18 | 88.62 | 89.00 | 35.22 | 29.68 | 64.01 | 46.00 | 15.01 | 45.49 |
| H2O | 37.26 | 30.43 | 21.07 | 25.33 | 17.01 | 63.00 | 41.98 | 84.77 | 52.00 | 31.56 | 22.92 | 59.91 | 31.00 | 6.04 | 37.45 |
| SparQ | 31.51 | 31.77 | 19.62 | 21.86 | 15.63 | 61.00 | 41.68 | 84.10 | 42.00 | 29.69 | 25.67 | 53.01 | 34.00 | 5.81 | 35.53 |
| MagicPig | 45.87 | 38.91 | **26.01** | 23.34 | 21.42 | 71.00 | 45.02 | 90.15 | 95.00 | 34.98 | 31.29 | 55.04 | 29.00 | 14.08 | 44.37 |

Table 1: LongBench accuracy of sparse methods across three models. Abbreviations: MQA-E (multifieldqa_en), MQA-Z (multifieldqa_zh), NarQA (narrativeqa), M-News (multi_news), Musiq (musique), TrivQA (triviaqa), P-Ret (passage_retrieval_en), Hotpot (hotpotqa), G-Rep (gov_report).

**Hardware.** Unless otherwise noted, experiments run on a single-node server with dual-socket AMD EPYC 7513 and 1.0 TiB system memory. We bind inference to 64 CPU cores. For GPU, we report two regimes: 1× NVIDIA A100 (single-GPU results) and 4× NVIDIA A100 on the same host (multi-GPU throughput). All methods (ours and baselines) are executed under the same software stack and runtime configuration; identical hardware is used across comparisons.

**Datasets** We use LongBench and LongBench v2. LongBench covers 14 datasets across six task categories (single-/multi-doc QA, summarization, few-shot, synthetic, code), with average lengths around 6.7k words (EN) and 13.4k characters (ZH). LongBench v2 expands the task set and context range (from ∼8k up to the ultra-long regime), emphasizing realistic multi-task retrieval and reasoning. We follow official protocols and task metrics (e.g., EM/F1/Acc for QA, ROUGE for summarization) and report per-task and macro-averaged scores.

**CSAttention** Unless otherwise stated we use $m=8$ subspaces, $C \in \{64, 128, 200\}$ query centroids per subspace, unit subspace weights $w_b=1$, and keep ∼5% tokens per step (final Top-K). We choose $L$ so that $mL$ saturates recall while keeping the reduce-by-key bounded on device; subspace $k$-means uses 10 iterations on GPU (cosine $k$-means with k-means++ seeding). We evaluate both execution backends: All-GPU (tables+KV resident on GPU) and CPU↔GPU (search/gather on CPU, transfer only Top-K KV to GPU).

## 4.2 PERFORMANCE

**Results on LongBench.** Table 1 reports per-task accuracy on LongBench for three backbones. On Llama3-8B, CSAttention's macro average (52.04) is within 0.7% loss of Full (52.41), with numerous per-task wins (e.g., MQA-E/Z, NarQA, Musique, TrivQA) and near-ties elsewhere. Qwen-8B shows virtually identical averages (52.25 vs. 52.30), again with CSAttention matching or exceeding Full on multiple tasks (e.g., Musique, TrivQA), and never incurring large degradations on any category. On Mistral-7B, CSAttention matches the Full average exactly (both 49.92) while leading or tying on several tasks (e.g., TrivQA, P-Ret, VCSum), indicating robustness across architectures. In contrast, PQCache—despite being tuned with 15 $k$-means iterations and SUBBITS= 8—and H2O/SparQ/MagicPig all trail CSAttention on the macro average and drop notably on harder retrieval/summarization tasks (e.g., M-News, Hotpot), consistent with their sensitivity to high sparsity.

| Method | Overall | Easy | Hard | Short | Medium | Long |
|---|---|---|---|---|---|---|
| Full | 31.0 | 35.4 | 28.3 | 37.2 | 26.0 | 30.6 |
| CSAttention | **31.2** | **34.4** | **29.3** | **37.8** | 25.1 | **32.4** |
| PQCache | 29.8 | 33.3 | 27.7 | **37.8** | 22.3 | 31.5 |
| $H_2O$ | 29.9 | 32.9 | 28.0 | 33.8 | 27.9 | 31.5 |
| SparQ | 26.2 | 27.6 | 25.4 | 30.0 | 22.3 | 27.8 |
| MagicPig | 29.2 | 29.5 | 29.0 | 31.8 | **26.9** | 29.4 |

Table 2: LongBench v2 evaluation results on Llama-3.1-8B.

**Results on LongBench v2.**   Table 2 presents the LongBench v2 accuracy results for Llama-3.1-8B. CSAttention achieves an overall score of 31.2, surpassing the dense Full attention baseline (31.0) and exceeding all sparse baselines. Notably, while keeping only ∼5% tokens, CSAttention maintains performance within statistical noise of the Full model on the global metric, whereas other sparse methods (PQCache, $H_2O$, SparQ, and MagicPig) exhibit more substantial drops (ranging from $-1.2$ to $-4.8$ points overall). These results demonstrate that our centroid-scoring token retrieval mechanism effectively preserves *near-full* model accuracy even at very high sparsity levels. Furthermore, CSAttention maintains strong performance on Easy and Short tasks (surpassing other sparse methods) while demonstrating particular strength on challenging Hard and Long tasks. It method improves Hard performance from 28.3 (Full) to 29.3 and Long from 30.6 (Full) to 32.4, suggesting enhanced capability for complex, long-context tasks.

**Stability at 95% sparsity.**   Across all three backbones, the gap between CSAttention and Full on the macro average is $\leq 0.37$ points, and exactly zero for Mistral-7B. Moreover, CSAttention's per-task variance is modest: it avoids catastrophic failures observed in some baselines (e.g., pronounced declines on M-News or cross-lingual QA). Combined with the length-bucket analysis, these results support the claim that *subspace partitaion + centroid-scoring in Q-space* maintains high recall of truly relevant keys under very high sparsity, delivering accuracy that is *indistinguishable from Full* in practice.

## 4.3 EFFICIENCY

**Schedules.**   We report three CSAttention schedules that trade sparsity and search frequency while maintaining *near-full* accuracy (cf. Appendix A.1): *0.05-step-1* keeps 5% tokens (95% sparsity) and searches every step; *0.20-step-8* keeps 20% tokens and searches every 8 steps; *0.15-step-4* keeps 15% tokens and searches every 4 steps.

**Decode throughput under CPU↔GPU mode.**   As illustrated in Figure 3 (Left), once the index is preloaded, CSAttention attains state-of-the-art decode throughput in the CPU↔GPU setting and the advantage *grows with context length*. Using the best CSAttention schedule at each length, speedups over baselines are:

- vs. PQCache: $2.95\times$ (8K), $4.35\times$ (16K), $5.60\times$ (32K), $9.40\times$ (64K), $8.26\times$ (128K).

- vs. MagicPig: $1.51\times$ (8K), $2.14\times$ (16K), $3.41\times$ (32K), $3.98\times$ (64K), $7.85\times$ (128K).

- vs. SparQ: $1.25\times$ (8K), $2.32\times$ (16K), $5.78\times$ (32K), $10.3\times$ (64K), $17.9\times$ (128K).

- vs. $H_2O$ (strong baseline), near parity at short lengths and consistent gains thereafter: $0.88\times$ (8K), $0.97\times$ (16K), $1.13\times$ (32K), $1.16\times$ (64K), $1.33\times$ (128K).

These results validate the intended deployment pattern of *offline prefill + online decode*: a single offline build enables substantially higher online throughput, and the gap widens with longer contexts because CSAttention's per-step work scales with fixed table sizes rather than total history.

**Decode throughput under All-GPU mode.**   Figure 3 (Right) compares Full attention with the three CSAttention schedules under an all-GPU backend. CSAttention consistently outperforms Full attention across all context lengths, with the performance advantage emerging early and growing substantially as sequence length increases: $1.16\times$ speedup at 8K, $1.22\times$ speedup at 16K, $1.81\times$ speedup at 32K, $3.31\times$ speedup at 64K, and $4.24\times$ speedup at 128K. The performance

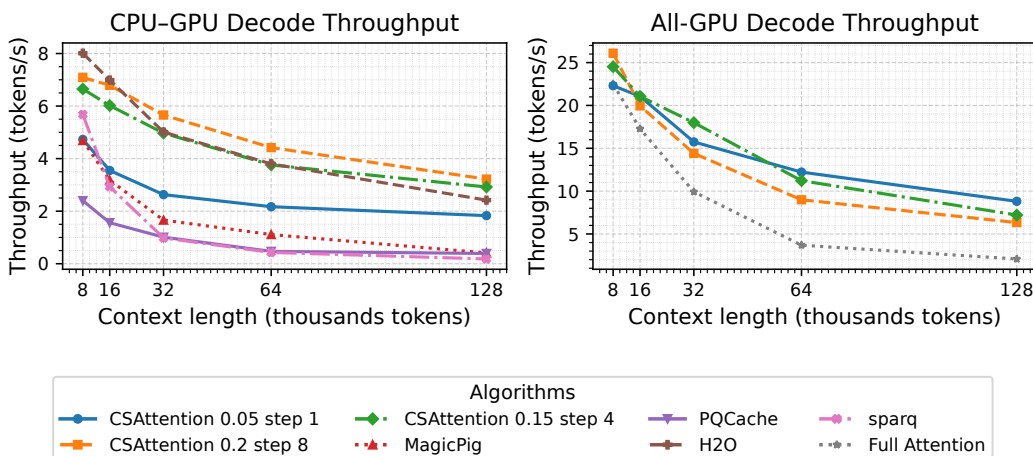

Figure 3: **Long-context decode efficiency.** Left: CPU↔GPU mode; Right: All-GPU mode.

gains arise from replacing $O(N)$ dense inner products with *fixed-size* list lookups and device-side union–reduce–Top-K kernels (cf. Section 3.3), whose cost is insensitive to history length.

**Step-level overhead and predictability.** To complement the throughput curves above, we report step-level overhead statistics for the two components affected by CSAttention (*search* and *append*; the attention kernel is unchanged and thus omitted here). For each schedule, we normalize the mean of (*append+search*) to 1.0 and scale P50/P90/P99 accordingly. As shown in Table 3, tails are tight across schedules (P90 $\approx$ 0.92–0.97; P99 $\approx$ 1.01), indicating bounded variability and predictable decode-time behavior. Empirically, the per-step composition is stable: **Attention : Search : Update** $\approx$ **1.0 : 0.3 : 0.1**. This matches the algorithmic design: per-step work touches at most $mL$ entries during the union–reduce and performs $O(1)$ try-insert for streaming updates, so the overhead remains insensitive to the total history length. These data explain *why* the gains in Fig. 3 strengthen with longer contexts and *why* lower search frequency schedules (e.g., 0.20-step-8) are preferable at short lengths.

Table 3: Normalized per-step overhead latency (mean of append+search normalized to 1.00).

| Schedule | P50 | P90 | P99 |
|---|---|---|---|
| **step 8** | 0.900 | 0.920 | 1.013 |
| **step 4** | 0.947 | 0.970 | 1.007 |
| **step 1** | 0.948 | 0.969 | 1.006 |

## 4.4 ROBUSTNESS UNDER LONG DECODE WITH CoT (LONGBENCH-V2)

**Setup and motivation.** To stress the *long-prefill, long-decode* regime we target, we evaluate CSAttention on LongBench-v2 with *chain-of-thought (CoT)* decoding. We append a simple CoT instruction (e.g., "think step by step") to each prompt and allow up to 2048 generated tokens before the final answer. Following LongBench-v2 protocols, only the final answer is scored under the official metrics. Operationally, the long system prompt is prefetched and indexed offline; decode then continues for hundreds or thousands of steps. The question is whether a gradually drifting query distribution during CoT degrades centroid-scoring retrieval.

**Engineering safeguards for robustness.** At decode time we use two lightweight mechanisms that keep work *bounded* while improving robustness: (i) a *recent-window passthrough* of size $R$ that always includes the most recent $R$ positions irrespective of index hits; and (ii) a *centroid backoff* that, when the best cosine similarity is low, selects the top-$\tau$ nearest centroids per subspace ($\tau \in \{2, 3\}$), merging at most $m\tau L$ entries prior to the reduce-by-key. We also allow small, thresholded bumps of

$R$ or $K$ in rare "off-manifold" cases. All mechanisms preserve fixed-size tables and do not change the asymptotic decode cost.

**Results.** In the long-context *offline prefill* setting, for each KV-head group (GQA) we construct centroids by treating the prefill $\mathbf{q}$-embeddings as a distribution over the session/domain. This distribution is already diverse due to the long system prompt, so nearest-centroid routing in $Q$-space remains stable during subsequent long decode. Empirically, Table 4 shows CSAttention remains *near-baseline* across difficulty buckets for both Llama-70B and Qwen-3 32B, indicating that prolonged CoT generation does not materially erode recall at high sparsity in this setting.

Table 4: **LongBench-v2 with CoT (max gen 2048)**: accuracy buckets under prolonged decode. "our" = CSAttention (default schedule).

| Model | Overall | Easy | Hard | Short | Med. | Long |
|---|---|---|---|---|---|---|
| Llama-70B (Full) | 36.5 | 38.6 | 35.2 | 46.0 | 33.0 | 27.6 |
| Llama-70B (our) | 36.4 | 39.1 | 34.8 | 44.0 | 35.4 | 25.9 |
| Qwen-3 32B (Full) | 49.2 | 53.1 | 46.8 | 60.0 | 41.1 | 47.2 |
| Qwen-3 32B (our) | 48.1 | 51.1 | 46.2 | 57.0 | 40.0 | 49.4 |

## 5 RELATED WORK

Attention serves as the core mechanism in Transformer models (Vaswani et al., 2017). Standard attention matrices exhibit inherent sparsity, wherein a large fraction of the computed weights are close to zero and can be pruned without significant impact on output quality (Zhang et al., 2025b). By exploiting this sparsity pattern, *sparse attention* methods achieve significant improvements in computational efficiency (Zhang et al., 2025c; Liu et al., 2024; Desai et al., 2024). Based on the mechanism for selecting attention tokens, sparse methods can be categorized into two types: *static methods*, which rely on a predefined sparsity pattern based on empirical observations to fix the computational tokens (Xiao et al., 2024b; Fu et al., 2025; Zhu et al., 2024; Xiao et al., 2025), and *dynamic methods*, which adaptively determine these tokens during decoding according to the real-time distribution of queries and keys (Zhang et al., 2023; Xiao et al., 2024a; Jiang et al., 2024; Ribar et al., 2024; Tang et al., 2024; Chen et al., 2024; Zhang et al., 2025a; Singhania et al., 2024). While static methods offer straightforward implementation, their fixed token selection patterns may lead to limitations in capturing long-range dependencies, as well as the potential loss of critical intermediate information (Hu et al., 2025; Tang et al., 2024).

Dynamic sparse methods have attracted much attention due to their flexibility and adaptability. Quest (Tang et al., 2024) and InfLLM (Xiao et al., 2024a) adopt a similar strategy: they partition the KV cache into blocks and generate a representative key vector for each block to facilitate efficient searching. SparQ (Ribar et al., 2024) and Loki (Singhania et al., 2024) estimate the Top-K most relevant keys for a given query by performing dimensionality reduction. H$_2$O (Zhang et al., 2023) maintains a fixed-size KV cache during decoding by dynamically evicting tokens. MagicPig (Chen et al., 2024), RetrievalAttention (Liu et al., 2024), HashAttention (Desai et al., 2024), and PQ-Cache (Zhang et al., 2025a) adopt vector search techniques—such as learning to hash, locality-sensitive hashing, and graph—to efficiently retrieve critical tokens. Our proposed CSAttention also falls into the category of dynamic sparse methods, exhibiting superior efficiency and effectiveness in LLM inference compared to existing techniques.

## 6 CONCLUSION

In this paper, we introduced Centroid-Scoring Attention (CSAttention), a training-free sparse attention method for efficient LLM inference. CSAttention ensures the reliable recovery of high-scoring keys under very high sparsity by mitigating the query-key distribution shift through subspace partitioning and query-centric table construction. Extensive experiments demonstrate that compared to state-of-the-art sparse attention methods, CSAttention maintains near-lossless model accuracy while achieving higher inference speed in high-sparsity and long-context scenarios, demonstrating its practical value for scalable long-context inference.

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

| | MQA-E | MQA-Z | NarQA | M-News | Musique | Trec | Samsum | TrivQA | P-Ret | Hotpot | G-Rep | LCC | LSHT | VCSum | Avg |
|---|---|---|---|---|---|---|---|---|---|---|---|---|---|---|---|
| Full | 55.54 | 62.87 | 29.91 | 27.16 | 30.89 | 72.50 | 43.75 | 91.65 | 100.00 | 56.16 | 35.26 | 64.89 | 46.00 | 17.16 | 52.41 |
| 0.05 step 1 | **56.02** | 62.01 | **30.46** | 26.38 | 31.11 | 71.50 | 44.16 | **91.95** | 99.50 | **55.94** | 33.60 | **63.33** | 45.00 | **17.63** | **52.04** |
| 0.05 step 2 | 53.98 | 60.91 | 28.99 | 26.05 | 30.09 | 70.50 | 42.18 | 90.05 | 97.50 | 54.03 | 31.56 | 63.21 | 45.00 | 17.32 | 50.81 |
| 0.15 step 4 | 54.97 | 62.04 | 30.28 | **26.95** | 30.72 | 71.00 | **44.43** | 91.89 | 99.00 | 55.64 | **33.84** | 63.12 | 44.50 | 16.78 | 51.80 |
| 0.20 step 8 | 54.99 | **62.50** | 29.84 | 26.41 | **31.43** | **72.00** | 43.98 | 91.87 | 98.50 | 55.51 | 33.57 | 62.88 | **45.50** | 17.23 | 51.87 |

Table 5: Task-level accuracy on LongBench for CSAttention schedules (Llama3 8B). Schedules are denoted as "keep ratio + search period": *0.05-step-1* keeps 5% tokens and searches every step; *0.15-step-4* keeps 15% and searches every 4 steps; etc. All schedules are accuracy-stable relative to Full; macro-average gaps are $\leq 0.6$ points.

# A  ADDITIONAL EXPERIMENTAL DETAILS AND RESULTS

## A.1  SCHEDULES EVALUATION

**Efficiency.** As shown in Figure 3 (Right, All-GPU mode), at 8K–16K context lengths, the lower search frequency of *0.20-step-8* yields the highest throughput. Beyond 32K, however, *0.15-step-4* and *0.05-step-1* become preferable as their stronger recall translates into better final throughput under the same token retention budget. Under the CPU↔GPU mode (cf. Figure 3 (Left)), less frequent searching reduces control traffic and amortizes host-side list merging along with PCIe H2D transfers of gathered KV (only the selected K=$\rho S$ vectors are moved). Consequently, schedules such as *0.15-step-4* and *0.20-step-8* deliver higher throughput at short to medium lengths, whereas *0.05-step-1* dominates at very long contexts by maximizing recall under a fixed token keep rate. Notably, all CSAttention scheduling variants maintain *near-lossless* accuracy compared to Full attention (cf. Table 5), allowing schedule selection to be driven primarily by throughput considerations for any given context length requirement.

**Performance.** Table 5 reports task-level accuracy on LongBench for several CSAttention schedules under identical model and data settings. Three observations emerge. *(i) Near-full accuracy at 95% sparsity.* *0.05-step-1* (keep 5%, search every step) attains an average of 52.04, within **0.37** points of Full (52.41), and tracks Full closely across QA, summarization, and retrieval tasks. *(ii) Infrequent search preserves accuracy.* Leveraging the empirical locality that consecutive tokens tend to share similar attention patterns, both *0.15-step-4* (keep 15%, search every 4 steps; 51.80) and *0.20-step-4* (keep 20%, search every 8 steps; 51.87) remain within $\leq 0.61$ points of Full on the macro average, with no catastrophic drops on any task. *(iii) Robustness vs. PQ-style retrieval.* Even at the most aggressive sparsity, *0.05-step-1* matches or exceeds the accuracy typically observed for PQCache-like methods at comparable keep budgets (cf. Section 4.2), reflecting the stability of subspace partition and query-space centroid-scoring.

## A.2  PREFILL LATENCY.

As illustrated in Figure 4, CSAttention invests additional computation during the prefill stage to construct its query-centric tables. This upfront cost, which is *offline* and incurred once per shared prompt and amortized over subsequent decoding steps, is a deliberate design trade-off to achieve the significant acceleration and robustness observed during online decoding (cf. Figure 3).

# B  COMPLEXITY ANALYSIS

**Notation (recap).** We use column vectors and $\langle \cdot, \cdot \rangle$ for inner products. Hidden size per head is $d$, partitioned into $m$ subspaces with sizes $\{d_b\}_{b=1}^m$ and $\sum_{b=1}^m d_b = d$. Sequence length is $N$ (number of cached keys/values so far). Each subspace has $C$ centroids; for centroid $(b, j)$ the offline list length is

$$L = \alpha N, \quad \alpha \in (0, 1).$$

At decode we keep K $= \rho N$ keys ($\rho \in (0, 1)$). Tables are $(\mathcal{I}_j^{(b)}, \mathcal{V}_j^{(b)})$ with indices in `int32` (4B) and scores in `fp16`/`bf16` (2B). Unless otherwise stated, complexities are *per head, per layer*; extension across layers/heads is linear. We denote by $I$ the number of mini-batch $k$-means iterations used for subspace clustering during prefill (we use $I=10$ by default on GPU).

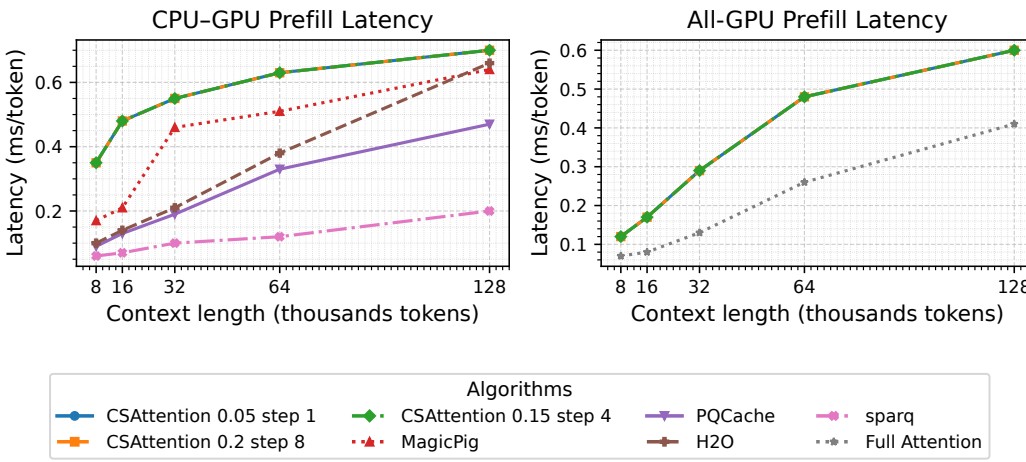

Figure 4: **Long-context prefill latency.** Left: CPU↔GPU mode; Right: All-GPU mode.

## B.1 PREFILL.

**KV cache baseline.** For one head in one layer, dense KV storage for length $N$ is

$$\text{KV\_bytes} = 2\, d\, \text{B} \cdot N,$$

where the factor 2 accounts for K and V, and $\text{B} \in \{2, 4\}$ is bytes per element (fp16/bf16 or fp32). Across layers the footprint scales linearly with $N$ and dominates device memory at long context.

**Index (tables) footprint with $L = \alpha N$.** Per head, our lists store $mC(\alpha N)$ entries (each 6B for `int32` index + `fp16` score), plus subspace centroids:

$$\underbrace{m\, C\, (\alpha N) \cdot 6}_{\text{lists (linear in } N)} + \underbrace{m\, C\, d_b \cdot 2}_{\text{centroids (independent of } N)} \qquad \text{bytes.}$$

Comparing the *linear-in-$N$* terms per head gives the ratio

$$\frac{\text{table bytes}}{\text{KV bytes}} = \frac{6\, mC\alpha}{2\, d\, \text{B}} = \begin{cases} \dfrac{3\, mC\alpha}{2\, d}, & \text{B} = 2 \text{ (fp16/bf16)}, \\ \dfrac{3\, mC\alpha}{4\, d}, & \text{B} = 4 \text{ (fp32).} \end{cases}$$

Thus the table's linear coefficient is controlled by $mC\alpha/d$. In the CPU↔GPU mode lists reside in DRAM; in the All-GPU mode feasibility is governed by HBM headroom (we choose $(C, \alpha)$ accordingly). Centroids contribute only $O(mCD)$ bytes and are small.

**Prefill time.** Per subspace, centroid→key scoring multiplies $K^{(b)} \in \mathbb{R}^{N \times d_b}$ with $(C^{(b)})^\top \in \mathbb{R}^{d_b \times C}$, costing $O(NCd_b)$, followed by Top-$L$ selection per column in $O(N)$ expected time (radix-/selection kernels). Summed over $m$:

$$T_{\text{prefill}} = O\big(I\, NCd\big) + O\big(NCd\big) + O\big(NCm\big),$$

dominated by subspace $k$-means ($I$ iterations). Serialization writes $O(mC\alpha N)$ list entries (linear in $N$). Prefill is a one-time cost per shared prompt and is amortized across requests.

## B.2 DECODE.

Let $U$ be the union of indices retrieved from the $m$ lists at a step; with $L = \alpha N$,

$$|U| \leq mL = m\alpha N.$$

**All-GPU complexity.** Per query (per head):

$$\begin{aligned}
\text{Nearest centroid (cosine GEMV)} &: \quad O(Cd), \\
\text{Concat/sort/segment-sum over } U &: \quad O(m\alpha N), \\
\text{Device Top-K on } U &: \quad O(m\alpha N) \text{ (radix-select)}, \\
\text{Sparse attention on K} = \rho N &: \quad O(\rho N\, d).
\end{aligned}$$

Hence

$$\boxed{T_{\text{all-GPU}}(N) = O(Cd) \; + \; O(m\alpha N) \; + \; O(\rho N\, d)}$$

Dense attention per step is $O(Nd)$. CSAttention replaces it by a linear form with a smaller coefficient, $((m\alpha) + (\rho d))/d$, provided $(\alpha, \rho)$ are moderate.

**CPU↔GPU complexity and bandwidth (asynchronous).** On the host, the bounded search is a linear merge/reduce over $m\alpha N$ elements:

$$T_{\text{CPU-search}} \;=\; O(m\alpha N).$$

Gather moves only the selected K $= \rho N$ keys/values to device; the *deterministic* H2D bytes per step (per head) are

$$\text{H2D\_bytes/step} \;=\; 2\,\rho N \cdot d \cdot \text{B}.$$

With search period $P > 1$ (reuse the index set for $P-1$ steps), the amortized host work and H2D shrink by $\approx 1/P$:

$$\overline{T}_{\text{CPU-search}} \approx O\!\Big(\tfrac{m\alpha}{P}N\Big), \qquad \overline{\text{H2D\_bytes/step}} \approx \tfrac{1}{P} \cdot 2\,\rho N d\, \text{B}.$$

Device-side attention remains $O(\rho Nd)$. Because CPU search + H2D for step $t+1$ overlap with GPU attention at step $t$ (streams/events, pinned buffers), PCIe latency is largely hidden; wall time is dominated by $O(\rho Nd)$.

**Why the union remains small in practice.** Although $|U| \leq m\alpha N$ grows linearly with $N$, the accumulator exploits $\langle q, k_i \rangle = \sum_{b=1}^{m} \langle q^{(b)}, k_i^{(b)} \rangle$. Truly aligned keys "collide" across subspaces and rise in the index-wise sum, so moderate $\alpha$ suffices to saturate recall (Sec. 3). Empirically, this keeps the *index-side* linear coefficient $m\alpha$ small, while the *attention-side* coefficient $\rho$ controls compute and transfer.

**Space/time takeaway under long context.** Online memory is dominated by the KV cache $O(Nd)$ per head per layer; CSAttention adds lists of size $O(mC\alpha N)$ (DRAM in CPU↔GPU; HBM in All-GPU) and $O(mCD)$ centroids (negligible). Per-step time is linear in $N$ but with reduced constants:

$$\text{dense: } O(Nd) \quad \Rightarrow \quad \text{CSAttention: } O\big(m\alpha N\big) + O\big(\rho Nd\big) \,(+\, O(Cd)).$$

With typical settings (e.g., $\rho \approx 0.05$, $\alpha \in [0.2, 0.4]$, $m$ small), decode cost and H2D traffic are substantially smaller than dense, and the one-time prefill/index overhead is amortized across many decode steps and many requests that share the prefill.

## APPENDIX D: HYPERPARAMETERS & INDEX SIZING

**Scope.** This appendix provides recommended defaults, tuning ranges, and sizing guidance for CSAttention. Unless noted otherwise, parameters are specified *per KV head*.

**Recommended defaults.** We recommend: subspaces $m=8$; centroids per subspace $C \in \{64, 128, 200\}$ (default 128; use 64 for tighter memory, 200 for ultra-long contexts); Top-$L$ uses proportional scaling $L = \alpha N$ (default $\alpha=0.25$, tunable $0.20 \sim 0.40$); keep ratio $\rho=0.05$ (Top-$K = \rho N$); uniform subspace weights $w_b=1$; recent passthrough window $R \in [16, 128]$ (default 32).

Table 6: Recommended defaults (per KV head) and tuning guidance.

| Parameter | Default / Guidance |
|---|---|
| Subspaces $m$ | 8 (use 6 if memory-bound; 12 for ultra-long contexts) |
| Centroids $C$ | 128 (64 for compact tables; 200 for very long $N$) |
| Top-$L$ | $L=\alpha N$, $\alpha=0.25$ (tune 0.20~0.40) |
| Keep $K$ | $K=\rho N$, $\rho=0.05$ (95% sparsity) |
| Recent window $R$ | 32 (increase to 128 for stronger recency bias) |
| Subspace weights $w_b$ | 1 (learned/confidence weights optional) |

**Sensitivity (what matters).** For fixed keep $K=\rho N$, throughput scales primarily with $mL$ (work of the union–reduce), while recall/accuracy saturate once $mL$ is large enough. In practice:

- $m \in \{6, 8, 12\}$: Increasing $m$ improves multi-subspace "collision" recall but grows table size and reduction work linearly; $m=8$ is a robust knee.
- $C \in \{64, 128, 200\}$: Larger $C$ improves centroid coverage, but returns diminish because decode selects only the nearest centroid per subspace; $C=128$ is a strong default.
- $\alpha$ controls $L=\alpha N$: higher $\alpha$ boosts recall but linearly increases union–reduce cost; $\alpha=0.25$ balances accuracy and speed.
- $\rho$ sets the final Top-$K$ density; it linearly affects attention FLOPs and (in CPU↔GPU mode) PCIe bytes per step. $\rho=0.05$ achieved near-lossless accuracy in our runs.

**Practical presets.** For medium lengths (8–32K) emphasizing throughput: $m=8$, $C=128$, $\alpha=0.20$, $\rho=0.10$. For ultra-long lengths (64–128K) emphasizing near-lossless accuracy: $m=8$, $C=200$, $\alpha=0.25\sim0.40$, $\rho=0.05$, $R=512\sim1024$.

## C  LLM USAGE

In the research process for this work, the authors used large language models (LLMs) for text polishing and writing assistance. After using these tools, the authors reviewed and edited the content as needed and take full responsibility for the publication.

