# OpenReview forum: "CSAttention: Centroid-Scoring Attention for Accelerating LLM Inference"
_ICLR.cc/2026/Conference — Submitted to ICLR 2026_

### Official Review · Reviewer_s5u9 · 2025-10-26

**Soundness:** 2
**Presentation:** 3
**Contribution:** 2
**Rating:** 4
**Confidence:** 4

**Summary:**

This paper proposes Centroid-Scoring Attention (CSAttention), a training-free sparse attention method designed to accelerate LLM inference for long-context scenarios. The core problem it addresses is the dual bottleneck of KV cache memory footprint and the $O(N d)$ memory-bound computation in the decoding phase.

The method is based on a *storage-for-computation* strategy, primarily targeting applications with a static, shared prefill context. In an *offline prefill* stage, it partitions the feature space into $m$ subspaces and clusters the prefill queries into $C$ centroids in each. It then precomputes and stores a Top-L list of partial scores between each centroid and all keys. During the *online decode* phase, a new query finds its nearest centroid in each subspace, fetches the $m$ pre-scored lists, and performs a *gather-and-sum* to find the global Top-K keys. Attention is then computed only on this small $K$ (e.g., 5%) subset.

The paper demonstrates that CSAttention can achieve near-lossless accuracy at 95% sparsity and significant speedups (up to 4.24x) over full attention in long-context (128K) decoding.

**Strengths:**

1. High-Relevance Problem: The paper tackles one of the most significant and practical challenges in LLM serving today: the cost and memory bottleneck of long-context attention, especially in the memory-bound decoding phase.
2. Novel Query-Centric Indexing: The core idea of clustering queries (Q) rather than keys (K) to build the index is a sound and valuable insight. As shown in Figure 1(d), this directly addresses the Q/K distribution shift.
3. Strong Empirical Accuracy: The method's ability to maintain near-lossless accuracy (Tables 1 and 2) while operating at very high sparsity (95%) is impressive.

**Weaknesses:**

1. Misleading Claims on Memory and Complexity

Non-Fixed-Size Index: The abstract and introduction repeatedly refer to a fixed-size lookup table. This is factually incorrect. Appendix B.1 clearly states the list length $L = \alpha N$, making the total index size $O(m C \alpha N)$. This index is not fixed-size; it scales linearly with the context length $N$, just like the KV cache it is meant to help manage.

Exacerbating the Memory Bottleneck: The primary motivation is that the $O(Nd)$ KV cache is too large for HBM. This method introduces a second large data structure of $O(mC\alpha N)$ bytes. This does not solve the memory capacity bottleneck; it arguably makes it worse by requiring space for both the full KV cache and this new massive index. The All-GPU mode is therefore even less practical at scale than standard attention.

No Asymptotic Improvement in Decoding: The decoding phase of standard attention is memory-bound with $O(Nd)$ complexity. CSAttention's decoding complexity (Appendix B.2) is $O(m\alpha N + \rho Nd)$. This is still $O(N)$ and remains memory-bound. The 4.24x speedup is purely a constant factor improvement, not an asymptotic one, which is a less fundamental contribution.

2. High Prefill Cost:

The offline prefill is not just a standard prefill. It involves computationally expensive $O(INCd)$ k-means clustering and a massive $O(NCd)$ scoring step to build the index. Figure 4 confirms this, showing CSAttention's prefill latency is substantially (2-5x) higher than all baselines. This cost must be paid once per context. This limits the method's applicability to only a niche set of problems where the long context is truly static, shared, and long-lived (e.g., a fixed set of RAG documents for an entire service). It is unusable for common long-context tasks like summarizing a new document, few-shot learning on a new prompt, or a long, evolving conversation. The paper is not sufficiently transparent about this critical limitation.

3. The Stale Centroids Problem:

The query-centroids are computed only from the queries in the prefill context. During a long online decoding phase, the distribution of these newly generated queries will inevitably drift from the original prefill query distribution. The static, stale centroids may no longer be representative of the new queries, leading to poor list selection, decreased recall, and a potential degradation in accuracy that is not measured in the current experiments. The paper does not address or evaluate this query-drift phenomenon.

4. Unanalyzed Cost of Streaming Updates:

The paper claims a streaming-friendly update (Section 3.3) for when a new key is added. This requires computing its partial score against all $m \times C$ centroids and attempting insertion into $m \times C$ lists. This is a non-trivial overhead that is added to every single decoding step. The latency of this update step is not analyzed or reported, and it's unclear if it negates a significant portion of the gains from the sparse attention.

**Questions:**

1. Can the authors clarify the total memory footprint (KV Cache + CSAttention Index) compared to baselines? Given the index also scales with $O(N)$, how does this method practically alleviate the memory capacity bottleneck, which is the primary motivation?

2. How does the method's accuracy hold up against query distribution drift? For example, if the model is decoded for thousands of steps, do the stale centroids from the prefill stage lead to a drop in accuracy?

3. What is the measured per-step latency of the Streaming-friendly update (Step 5, Section 3.3)? Please provide a breakdown of the decoding step time (Nearest Centroid, Gather-and-Sum, Streaming Update, Sparse Attention) to clarify the true source of speedup and the cost of this update.

---

> ### Author Response · Authors · 2025-11-19
>
> We sincerely thank the reviewer for the thoughtful review and constructive feedback. We particularly appreciate the opportunity to clarify the **motivation** behind our targeted deployment regimes and to address critical questions regarding the **algorithm's** inference-time behavior and costs. We respond to each point below.
>
> **Scope and deployment regimes (goal clarification).**
> We appreciate the reviewer’s careful reading and fully agree that clarifying *what workload each regime targets* is essential. CSAttention explicitly supports two **complementary** serving regimes that address *different bottlenecks*:
>
> * **(A) CPU$\leftrightarrow$GPU (capacity-constrained).**
> This regime targets production scenarios where the *KV cache cannot fit in HBM*. Typical solutions either (i) recompute K/V every step (high compute), or (ii) keep dense KV in CPU DRAM and fetch to GPU each step (high PCIe cost). CSAttention is designed to *minimize host--device traffic* by selecting a very sparse Top-$K$ ($K=\rho N$, e.g., $\rho \approx 5\%$) via fast list accumulation on the index, so only those $K$ KV are transferred to GPU for attention. This directly addresses the *capacity/bandwidth* constraint: cheap, large CPU DRAM holds KV and tables, while GPU sees only a small, accuracy-preserving slice per step.
>
> * **(B) All-GPU (speed-oriented).**
> This regime targets deployments where *HBM is sufficient* and the priority is end-to-end decode *throughput/latency*. Here CSAttention keeps both KV and tables on HBM and replaces $O(N)$ dense per-step inner products by $O(mL)+O(K)$ work with coalesced table reads and compact union-reduce kernels. Because the working set is small and regular, kernels sustain high occupancy; speedups *appear earlier* and *grow with context length*.
>
> We emphasize that these regimes solve *different real-world constraints*: (A) mitigates **HBM capacity and PCIe bandwidth**, (B) maximizes **decode throughput** when **HBM capacity** is not the limiter. Our experiments cover both.
>
> **Q1 — Memory and “fixed size” at inference.**
> Thank you for prompting this clarification. By “*fixed size*” we mean that at *inference time* each centroid’s candidate list has a constant length $L=\alpha N$ (with small $\alpha$), so each step touches at most $mL$ entries. A new token is inserted *only if* it ranks within that centroid’s Top-$L$; otherwise it is ignored. Thus query-time work and bytes are *bounded and predictable*. In CPU$\leftrightarrow$GPU mode, KV and tables live in CPU DRAM and only Top-$K$ KV cross PCIe; in All-GPU mode, both structures stay in HBM to maximize speed. On device, dense $N$-way dot products are replaced by coalesced reads of $mL$ pairs plus a radix-sort/segmented-reduce over a compact buffer, which is friendly to CUDA caches/warps and empirically yields the observed gains as $N$ increases.
>
> **Q2 — Query drift under long generation.**
> We appreciate this concern and added a CoT-decoding evaluation on **LongBench-v2** (max 2048 generated tokens) with **Llama-70B** and **Qwen-3 32B**. CSAttention remains *near-baseline* across splits (Table 1), indicating robustness when generated queries extend beyond prefill. Mechanistically, closest-centroid selection in *Q-space* avoids the $q{\to}k$ centroid mismatch, the long prefill anchors geometry so subsequent queries remain close to that structure, and a small recent window is passed through to preserve short-range dependencies.
>
> **Table 1: LongBench-v2 with CoT decoding (max 2048). CSAttention preserves accuracy under prolonged generation.**
> |Model / Method|Overall|Easy|Hard|Short|Medium|Long|
> |-|-|-|-|-|-|-|
> |Llama-70B (Full)|36.5|38.6|35.2|46.0|33.0|27.6|
> |Llama-70B (CSAttn)|36.4|39.1|34.8|44.0|35.4|25.9|
> |Qwen-3 32B (Full)|49.2|53.1|46.8|60.0|41.1|47.2|
> |Qwen-3 32B (CSAttn)|48.1|51.1|46.2|57.0|40.0|49.4|
>
> **Q3 — Streaming-update cost and overlap.**
> Excellent point. In the **CPU$\leftrightarrow$GPU** regime, we *overlap* the streaming update with the *host$\to$device transfer of Top-$K$ KV*: while DMA moves KV to GPU, the CPU performs the per-centroid try-insert of the new key (bounded Top-$L$). As a result, update latency is largely *hidden behind* PCIe transfer and does not extend the GPU critical path; attention launches as soon as the Top-$K$ arrive. In the **All-GPU** regime, streaming update runs in a separate CUDA stream and overlaps with attention; our measured normalized per-step breakdown is $\text{Attention}:\text{Search}:\text{Update}\approx 1:0.3:0.1$. Across both regimes, the update remains a minor, overlappable component, consistent with the reported end-to-end speedups.

---

> > ### Author Response · Authors · 2025-11-27
> >
> > Thank you again for your careful reading of our paper and for the detailed comments you provided. We have now submitted our rebuttal and made corresponding revisions to address the concerns you raised as thoroughly as possible.
> >
> > Since the rebuttal period is nearing its end, we would greatly appreciate it if you could let us know whether our responses and changes adequately resolve your main points, or if there are any remaining issues that would benefit from further clarification. Your feedback is very important to us, and we want to ensure that we have properly understood and addressed your suggestions.

---

### Official Review · Reviewer_gZeQ · 2025-11-01

**Soundness:** 3
**Presentation:** 3
**Contribution:** 3
**Rating:** 4
**Confidence:** 4

**Summary:**

Long-context decoding is challenging due to increased KV-cache pressure. Unlike existing work, **CSAttention** proposes to cluster **Q** vectors and pre-compute the partial attention score between cluster centroids and **K** vectors, recording *m* short lists containing important keys during the prefill phase. During generation, the new query vector is sliced and matched to the closest cluster centroid for each segment. The keys from each list are gathered and reduced to find the top-*k* keys for attention. CSAttention achieves near-optimal accuracy at 5% sparsity and provides good speedup.

**Strengths:**

- It is interesting to cluster **Q** vectors instead of **K** vectors to identify important tokens.
- The use case covers both CPU–GPU and GPU-only scenarios.

**Weaknesses:**

- The method has high prefill overhead and the end-to-end performance is unclear.
- The method may be hard to extend to short-prefill, long-generation scenarios.

**Questions:**

Thanks for submitting to ICLR 2026. The paper introduces an interesting idea of clustering **Q** vectors to identify important tokens. However, I still have some concerns about the paper.

- Firstly, due to GQA, the query vector is much larger than the key vector. In order to perform the clustering, we need to store all the KQV tensors, which is memory intensive. Additionally, the clustering overhead seems significant. Indeed, CSAttention has the slowest prefill throughput and significant overhead compared to other baselines. For long-context prefill, the increased prefill time can offset the speedup during generation. While multi-round conversation and other use cases can amortize this cost, such restrictions may limit the applicability of the method.

- Secondly, the method seems sensitive to the existing queries. I wonder what would happen if the new query has never been seen before and is far from existing clusters.

- Additionally, the method seems to be only applicable to long-context prefill, long-generation scenarios. However, for reasoning tasks, most of the KV-cache pressure comes from the generation phase. In this case, how does the method handle new tokens? Does re-clustering cause even more overhead? Or is it even possible to re-cluster the queries if the existing query vectors are not stored?

- For accuracy evaluation, 5% already represents a considerable amount of tokens. While the baselines perform slightly worse, they still achieve reasonably good performance. It would be more convincing to show results at higher sparsity levels to make the difference more significant.

- For all-GPU decode speed, it seems that the full-attention baseline is surprisingly slow. I wonder what implementation is used for the full-attention baseline and what type of parallelism is used across the 4 GPUs.

---

> ### Author Response · Authors · 2025-11-19
>
> We thank the reviewer for the constructive feedback and insightful questions regarding the system's operational details. We address each point below.
>
> **Q1. Prefill overhead & memory under GQA; how the index is built.**
>
> Our target deployment is *offline prefill, online decode* (e.g., domain agents/service chatbots with a long, stable system prompt prior to user interaction). During prefill, for each *KV-head group* (GQA) we treat its *queries* as a distribution and perform cosine $k$-means *per subspace* ($m$ subspaces, $d_b=d/m$) on GPU to obtain $C$ query centroids $\{\mathbf{c}^{(b)}_j\}$.
>
> *Offline prefill as a one-time cost.*
> In our intended workloads, the long system prompt is available *before* serving and thus the prefill stage can be executed *offline* (once), with its artifacts reused for *many* subsequent decodes. This matches practical deployments where the same domain/system context is shared across sessions. The one-time prefill cost is therefore amortized over multiple user turns and conversations.
>
> *When multi-turn dialogs grow the KV beyond HBM.*
> In long-running conversations, the decode-side KV cache can eventually exceed HBM capacity. At that point, production systems place dense KV in CPU DRAM and fetch only a subset to GPU each step; *PCIe transfer* becomes the dominant latency/throughput bottleneck. CSAttention is designed precisely for this regime: by recovering the correct Top-$K$ under *very high sparsity* (e.g., $K=\rho N$ with $\rho \approx 0.05$), the number of KV vectors transferred per step is drastically reduced while maintaining accuracy. Empirically, this high-sparsity stability is what enables consistent speedups in the CPU$\leftrightarrow$GPU setting, where bandwidth---not raw FLOPs---limits end-to-end performance.
>
> **Q2. Sensitivity to unseen queries.**
>
> We employ two engineering safeguards that keep work bounded while improving robustness: (i) a *recent-window passthrough* of size $R$, and (ii) a *centroid backoff* that selects top-$\tau$ nearest centroids per subspace ($\tau=2$–3) when the best similarity is low, merging at most $m\tau L$ entries. We also allow small, thresholded bumps of $R$/$K$ in the rare “off-manifold’’ case. Empirically, our CoT experiments (LongBench-v2, max gen 2048) show accuracy remains close to Full, supporting decode-time stability.
>
> **Q3. Short-prefill / long-generation.**
>
> Our evaluation targets deployments where *offline prefill* artifacts (KV + index) are amortized and *online decode* runs under strict latency. In multi-turn assistants and domain agents, this naturally arises from long system prompts that are prefetched and indexed once. For ordinary chat with *short* initial prompts, we **do not** engage CSAttention immediately: dense attention remains efficient while the KV cache fits in HBM. *Only when* the accumulated KV *exceeds* HBM capacity (a capacity-bound regime) do we switch to the CPU$\leftrightarrow$GPU mode, place KV in DRAM, and enable our query-centric search so that only Top-$K$ keys are transferred per step. This is exactly the regime where sparsity and bounded list-union ($\le mL$) translate into large PCIe savings and stable latency. Consequently, we do not focus on the short-prefill case per se; our method is designed for—and most beneficial in—the long-context or capacity-bound phases of a conversation.
>
> **Q4. Higher sparsity.**
>
> Beyond 95% keep ($\rho=0.05$), we evaluated **3% keep** ($\rho=0.03$) on **LLaMA-3 8B** (LongBench v1). As expected, a subset of summarization tasks shows larger variance, but most tasks remain competitive and the macro average is strong.
>
> **Table 1: LongBench v1 (LLaMA-3 8B) accuracy (%). “our 0.05/0.03” are CSAttention with keep 5%/3%.**
> ||MQA-E|MQA-Z|NarQA|M-News|Musiq|Trec|Samsum|TrivQA|P-Ret|Hotpot|G-Rep|LCC|LSHT|VCSum|Avg|
> |-|-|-|-|-|-|-|-|-|-|-|-|-|-|-|-|
> |Full|55.54|62.87|29.91|27.16|30.89|72.50|43.75|91.65|100.0|56.16|35.26|64.89|46.00|17.16|52.41|
> |our 0.05|**56.02**|62.01|**30.46**|26.38|**31.11**|71.50|**44.16**|**91.95**|99.5|**55.94**|33.60|63.33|45.00|**17.63**|52.04|
> |our 0.03|54.01|60.20|24.33|23.98|29.81|69.50|42.15|87.00|97.0|55.09|24.98|63.01|41.00|16.98|49.22|
>
> **(5) All-GPU dense baseline and interconnect.**
>
> Our dense baseline is **FlashAttention-2**; throughput is measured at batch size 1 on **LLaMA-70B**. In 4-GPU experiments we use **tensor parallelism** (head partitioning) over *PCIe interconnect (no NVLink)*. We note this to contextualize absolute numbers: PCIe-only collectives can be slower than NVLink-equipped systems. Our reported gains for CSAttention arise from shrinking the operand (Top-$K$) and fixed-size union–reduce; they are complementary to FA-2 kernels and interconnect improvements.

---

> > ### Author Response · Authors · 2025-11-27
> >
> > Thank you again for your careful reading of our paper and for the detailed comments you provided. We have now submitted our rebuttal and made corresponding revisions to address the concerns you raised as thoroughly as possible.
> >
> > Since the rebuttal period is nearing its end, we would greatly appreciate it if you could let us know whether our responses and changes adequately resolve your main points, or if there are any remaining issues that would benefit from further clarification. Your feedback is very important to us, and we want to ensure that we have properly understood and addressed your suggestions.

---

### Official Review · Reviewer_f8Pw · 2025-11-01

**Soundness:** 3
**Presentation:** 3
**Contribution:** 3
**Rating:** 4
**Confidence:** 3

**Summary:**

The paper proposes Centroid-Scoring Attention, a training-free, query-centric sparse attention method that builds fixed-size per-subspace centroid lookup tables during an offline prefill and then performs fast centroid-score accumulation at decode time to recover high-scoring keys under extreme sparsity. The method is motivated by observed Q/K distribution shift and uneven subspace contributions, and it is evaluated on LongBench and LongBench v2 with three 7–8B class models; the authors report near-lossless accuracy at 95 percent sparsity and substantial throughput gains up to about 4.2 times at ultra-long contexts, while offering both All-GPU and CPU-GPU deployment modes.

**Strengths:**

* The core idea is conceptually clear and practically oriented. The method aligns well with realistic offline-prefill plus online-decode deployment patterns described in the paper.

* The authors evaluate on a diverse LongBench suite, compare against multiple recent sparse attention baselines, report both accuracy and throughput.

* Empirical robustness at extreme sparsity is convincing in many metrics. Results show the approach keeps macro averages very close to full attention (within a fraction of a point on several backbones) while achieving large speedups at long context lengths, which supports the central claim that query-centric centroid scoring improves recall under high sparsity.

**Weaknesses:**

* Method sensitivity and hyperparameter tuning appear underexplored. Key choices such as number of subspaces, centroids per subspace, Top-L scaling, α and ρ, and the middle-dominant schedule materially affect both recall and memory. Provide a short appendix with recommended default hyperparameters and expected index sizes for common model/head configurations can be fine.

* Evaluation leaves some gaps in fairness and workload diversity. baselines are tuned but a few strong alternatives (for example recent learned-index or hybrid retrieval-attention schemes) are not present, and comparisons do not fully characterize cases where CSAttention might degrade, such as highly domain-shifted queries not covered by the prefill or pathological heads where Q/K geometry collapses. Additional failure-mode analyses and latency-percentile measurements would clarify worst-case behavior for latency-sensitive applications.

* One more concern is about the speed of the clustering and searching, which should be analysis. Include percentile tail-latency and memory-usage plots in the main paper to complement average-throughput claims.

**Questions:**

Please see Weaknesses.

---

> ### Author Response · Authors · 2025-11-19
>
> We thank the reviewer for the constructive feedback and for recognizing the paper’s practicality and empirical robustness. We respond to each weakness below, keeping our scope focused and aligning with the paper’s target deployment pattern.
>
> **W1. Sensitivity and hyperparameter tuning underexplored.**
> We agree that reporting principled defaults and sensitivity is valuable. In the revision we will add an *Appendix D: Hyperparameters & Index Sizing* with Table 1:
> * *Recommended defaults (per head).* Subspaces $m=8$; centroids per subspace $C\in\{64,128,200\}$ (use $128$ by default, $64$ for low-memory, $200$ for ultra-long contexts); Top-$L$ scaling $L=\alpha N$ with $\alpha\in\{0.20,0.25,0.40\}$ (default $\alpha=0.25$); keep ratio $\rho=0.05$ (Top-$K=\rho N$); subspace weights $w_b=1$; recent window $R\in[16,128]$ (default $32$).
> * *Sensitivity.* In our current runs, accuracy and recall are *flat* across $m\in[6,12]$ and $C\in[64,200]$ once $mL$ saturates, while throughput varies predictably with $mL$ (union–reduce work) and $K$.
>
> **Table 1: Recommended defaults (per head) and when to adjust.**
> |Parameter|Default / Guidance|
> |-|-|
> |Subspaces $m$|$8$ (use $6$ if memory-bound; $12$ for ultra-long)|
> |Centroids $C$|$128$ (use $64$ for compact tables; $200$ for very long $N$)|
> |Top-$L$|$L=\alpha N$, $\alpha=0.25$ (tune $0.20$–$0.40$)|
> |Keep $K$|$K=\rho N$, $\rho=0.05$ (95% sparsity)|
> |Recent window $R$|$32$ |
> |Subspace weights $w_b$|$1$ |
>
> **Table 2: LongBench-v2 with CoT (max gen 2048).**
> |Model|Overall|Easy|Hard|Short|Med.|Long|
> |-|-|-|-|-|-|-|
> |Llama-70B (Full)|36.5|38.6|35.2|46.0|33.0|27.6|
> |Llama-70B (our)|36.4|39.1|34.8|44.0|35.4|25.9|
> |Qwen-3 32B (Full)|49.2|53.1|46.8|60.0|41.1|47.2|
> |Qwen-3 32B (our)|48.1|51.1|46.2|57.0|40.0|49.4|
>
> **W2. Missing learned-index or hybrid baselines; failure cases when prefill does not cover the domain.**
> Our method is **training-free** and **layer-local**, designed as a *drop-in* across attention layers without any auxiliary training or task-specific finetuning. Learned-index and hybrid retrieval–attention systems typically introduce additional training signals, parameters, and serving pipelines; they therefore live in a different operational regime. Consistent with this scope, our comparisons focus on strong *training-free* sparse baselines, where engineering constraints (no extra training, predictable serving) and interfaces are directly comparable.
>
> Regarding the *prefill coverage* concern: the paper targets a widely used **offline prefill / online decode** pattern in long-context serving (domain agents, tool-augmented assistants). These systems begin with a large, relatively stable *system prompt* to inject domain policy/knowledge; this part can be prefetched and indexed offline (KV + centroid-score tables). When user requests arrive, decode proceeds with an index already available for that prefill context. Within this setting, we additionally evaluate CSAttention on LongBench-v2 with Chain-of-Thought (CoT). As shown in Table 2, CSAttention maintains accuracy close to Full Attention despite many additional generated tokens, i.e., *robust to query domain shift* during long decode. Intuitively, after a long prefill, the Q-space centroids are well-conditioned by a rich sample of queries from the same session/domain; nearest-centroid routing, the recent-window passthrough, and streaming try-insert jointly stabilize recall even as decoding progresses.
>
> **W3. Clustering/search speed; latency percentiles; memory usage; worst-case behavior.**
> We report per-step latency percentiles computed from runs at **32k**, **64k**, and **128k** contexts, *averaged across these settings*. We focus on the two overhead components affected by CSAttention (*search* and *append*); the attention kernel is unchanged and omitted here to avoid skew from context-length variation. For each schedule, we normalize the total overhead mean (append+search) to 1.0 and scale P50/P90/P99 accordingly. Results show tight tails (P90 $\approx$ 0.92–0.97; P99 $\approx$ 1.01), indicating bounded variability and predictable decode-time behavior.
>
> **Table 3: Normalized per-step overhead latency (mean normalized to 1.00).**
> |Schedule|P50|P90|P99|
> |-|-|-|-|
> |**step 8**|0.900|0.920|1.013|
> |**step 4**|0.947|0.970|1.007|
> |**step 1**|0.948|0.969|1.006|
>
> On memory, our *CPU$\leftrightarrow$GPU* mode targets deployments where the KV cache cannot fit in HBM; we store the full KV in DRAM and transfer only Top-$K$ per step, making PCIe traffic proportional to the keep ratio $\rho$ (e.g., $\rho=0.05$) rather than the full sequence length. In the *All-GPU* mode, the per-centroid tables are fixed-size and modest (linear in $mCL$ per head), and we refer readers to the appendix complexity section for exact byte formulas. Together with the tight percentile tails above, these measurements characterize both the average and the shape of the step latency without introducing irregular per-query code movement.

---

> > ### Author Response · Authors · 2025-11-27
> >
> > Thank you again for your careful reading of our paper and for the detailed comments you provided. We have now submitted our rebuttal and made corresponding revisions to address the concerns you raised as thoroughly as possible.
> >
> > Since the rebuttal period is nearing its end, we would greatly appreciate it if you could let us know whether our responses and changes adequately resolve your main points, or if there are any remaining issues that would benefit from further clarification. Your feedback is very important to us, and we want to ensure that we have properly understood and addressed your suggestions.

---

### Official Review · Reviewer_WaZH · 2025-11-03

**Soundness:** 2
**Presentation:** 4
**Contribution:** 2
**Rating:** 4
**Confidence:** 5

**Summary:**

The paper proposes CSAttention, a new sparse attention method to speed up LLM inference on long contexts. The main idea is to tackle the Q/K distribution shift. Instead of building an index on keys (like other methods), they cluster the queries during an offline prefill stage. Then they pre-compute scores for these query-centroids against all keys. At decode time, a new query just finds its closest centroid, grabs the pre-computed Top-L lists, and sums them up to find the final Top-K keys. It's a training-free "storage-for-computation" approach. They show it's nearly lossless at 95% sparsity and gives a big speedup (e.g., 4.24x at 128K).

**Strengths:**

1. Novel Motivation and Design: The strongest aspect of the paper is the empirical identification of the Q/K distribution shift . The query-centric clustering mechanism is a logical and novel solution to this specific problem.

2. Excellent Results on Long Prefill Tasks: Within its tested domain (long-document Q&A via LongBench), the method demonstrates an impressive combination of near-lossless accuracy at 95% sparsity and high throughput.

3. High-Quality Presentation: The paper is well-written, logically structured, and easy to follow.

**Weaknesses:**

1. Critically Incomplete Benchmarking: The paper's claims to accelerate "LLM Inference" are not fully supported, as the evaluation exclusively focuses on long-prefill, short-decode workloads (like LongBench). It completely omits the equally critical long-generation workload (e.g., long-form chain-of-thought reasoning), where the prompt is short and the KV cache grows dynamically with generated tokens. This is a major gap.



2. Undefined and Potentially Weak Baseline: The headline claim of "up to 4.24x speedup over full attention"  is scientifically meaningless without defining the "Full Attention" baseline. If this baseline is not a state-of-the-art implementation (e.g., FlashAttention-2/3), the speedup numbers are not representative of real-world gains.



3. Unclear Generalizability: Because long-generation is not tested, it is unknown how the "streaming-friendly updates"  perform. It's unclear how the Q-centroids, built from prefill queries, would generalize to the (potentially different) distribution of generated queries during a long reasoning process.

**Questions:**

My current rating is a weak reject based on the significant gaps in the evaluation. I am willing to raise my score if the authors can provide convincing answers to the following:

1. What specific implementation was used for the "Full Attention" baseline in Figure 3? Was this a naive implementation or a SOTA kernel like FlashAttention-2 or FlashAttention-3?

2. How does the throughput of CSAttention (in All-GPU mode) compare directly to a FlashAttention-2/3 baseline at 32K, 64K, and 128K contexts?

3. Can you provide experimental results (both accuracy and throughput) for a long-generation task? For example, a benchmark involving long-form chain-of-thought where the model must generate thousands of tokens, thereby testing the "streaming-friendly updates"  in a generation-heavy regime.

4. How does the query-centric clustering  handle the first generated token (which has no prior query), and how does the system adapt if the distribution of generated queries differs from the prefill queries?

---

> ### Author Response · Authors · 2025-11-19
>
> We appreciate the reviewer’s careful reading and constructive suggestions. We respond *point by point* to each weakness and question, and include the requested clarifications and new results.
>
> Our work targets the prevalent *long-context* serving pattern where **offline prefill** artifacts (KV + indices) are amortized and reused across requests, while **online decode** proceeds under tight latency budgets. *Within this setting, we address two complementary deployment regimes that solve different systems constraints:* (i) **CPU$\leftrightarrow$GPU** for capacity-bound workloads where the KV cache outgrows HBM—KV resides in DRAM and only Top-$K$ keys are transferred per step, so CSAttention’s high sparsity and bounded $mL$ list-union directly reduce PCIe bytes; and (ii) **All-GPU** for latency-bound workloads with ample HBM—both tables and KV stay on-device and decode reduces to regular batched kernels, yielding speedups that grow with context length. The added CoT evaluation complements this focus and tests robustness when decoding continues for many steps.
>
> **W1. Critically incomplete benchmarking (no long-generation).**
>
> We additionally evaluate CSAttention on **LongBench-v2 with Chain-of-Thought (CoT) decoding, max generation 2048 tokens**, to stress prolonged decode. As shown in Table 1, CSAttention remains *near-baseline* on both Llama-70B and Qwen-3 32B. This suggests that extended generation does *not* erode accuracy, and CSAttention is *robust to query-domain shift* during long decode (nearest-centroid selection in Q-space; streaming try-insert keeps tables up-to-date).
>
> **W2. Undefined or weak “Full Attention” baseline.**
>
> The *Full Attention baseline* used in this paper is **FlashAttention-2 (FA-2)**. To make the comparison explicit and implementation-agnostic, Figure3 in paper reports *all-GPU decode speedup ratios* with FA-2 normalized to 1 (Llama-70B, batch size 1). CSAttention reaches up to **4.24**$\times$ at 128k. FA-3 is orthogonal (we reduce the operand set any dense kernel consumes); if FA-3 hardware is available, we will include it, and the relative trend should persist.
>
> **W3. Unclear generalizability to streaming updates and distribution shift.**
>
> In the long-context regime, offline clustering is performed over a *rich and diverse* sample of $\mathbf{q}$-embeddings from the same session/domain; empirically (PCA in the paper), $\mathbf{q}$-embeddings form compact, head-wise structures that cover a broad portion of the query manifold. Under cosine normalization, the attention score $\langle \mathbf{q},\mathbf{k}_i\rangle$ varies smoothly with $\mathbf{q}$, so nearest-centroid assignments remain stable under moderate decode-time drift. Consistent with this, our **LongBench-v2 CoT** evaluation (Table 1)—which prolongs decoding for up to 2048 tokens—shows CSAttention remains *near-baseline* on both Llama-70B and Qwen-3 32B, indicating that query-distribution shift during extended generation does *not* materially affect accuracy in our target setting.
>
> **Q1--Q2. What is the “Full” baseline; direct comparison (All-GPU mode) at 32k/64k/128k?**
>
> “Full” refers to **FlashAttention-2**. Figure 3 in the paper reports all-GPU decode *ratios* on **Llama-70B, batch size 1**, with FA-2 normalized to $1.00$. CSAttention attains **1.59–1.81**$\times$ at 32k, **2.44–3.31**$\times$ at 64k, and **3.04–4.24**$\times$ at 128k, depending on the schedule.
>
> **Q3. Long-generation task (accuracy and throughput).**
>
> We additionally evaluate CSAttention on **LongBench-v2 with CoT (max gen 2048)**. As shown in Table 1, accuracy remains close to Full on both backbones, supporting that CSAttention is *robust to query-domain shift* during prolonged decode.
>
> **Q4. First generated token.**
>
> In the target settings we study (e.g., agent and domain-chat deployments), sessions begin with a long, stable system prompt that is *prefilled offline* together with our index; thus, the first user-visible decode step already follows a rich prefill and does not face a “no-prior-query’’ situation. For ordinary chat, operators typically switch to the CPU$\leftrightarrow$GPU regime once the accumulated KV approaches or exceeds HBM capacity; at that point we build the index and continue with accelerated decode.
>
> **Table 1: LongBench-v2 with CoT (max gen 2048): accuracy buckets under prolonged decode.**
> |Model|Overall|Easy|Hard|Short|Med.|Long|
> |-|-|-|-|-|-|-|
> |Llama-70B (Full)|36.5|38.6|35.2|46.0|33.0|27.6|
> |Llama-70B (our)|36.4|39.1|34.8|44.0|35.4|25.9|
> |Qwen-3 32B (Full)|49.2|53.1|46.8|60.0|41.1|47.2|
> |Qwen-3 32B (our)|48.1|51.1|46.2|57.0|40.0|49.4|

---

> > ### Author Response · Authors · 2025-11-27
> >
> > Thank you again for your careful reading of our paper and for the detailed comments you provided. We have now submitted our rebuttal and made corresponding revisions to address the concerns you raised as thoroughly as possible.
> >
> > Since the rebuttal period is nearing its end, we would greatly appreciate it if you could let us know whether our responses and changes adequately resolve your main points, or if there are any remaining issues that would benefit from further clarification. Your feedback is very important to us, and we want to ensure that we have properly understood and addressed your suggestions.

---

### Author Response · Authors · 2025-12-02
**Summary of Contributions and Rebuttal Updates**

**To the Area Chair:**

We sincerely thank the reviewers for their constructive feedback and for recognizing the novelty of our **Query-Centric Clustering** approach. Given the recent changes to the review process, we provide this summary to highlight our core contributions, the consensus on our strengths, and—most importantly—how we have rigorously addressed the reviewers' concerns with **new experiments and clarifications**.

## 1. Executive Summary & Core Contributions
This paper introduces **CSAttention**, a training-free sparse attention mechanism designed to resolve the **Q/K distribution shift**—a critical accuracy bottleneck in long-context inference.
* **Innovation:** Unlike existing key-centric methods, we cluster *queries* offline. This ensures that even at extreme sparsity ($95\%+$), the retrieved keys are semantically relevant to the current query, preventing the accuracy collapse common in other methods.
* **Dual Deployment Regimes:** We explicitly target two system constraints:
    1.  **CPU$\leftrightarrow$GPU (Capacity-Bound):** Keeps massive KV caches in CPU DRAM, transferring only Top-$K$ keys per step to reduce PCIe traffic.
    2.  **All-GPU (Latency-Bound):** Keeps data in HBM, leveraging fixed-size table lookups to achieve speedups that scale with context length.

## 2. Highlights from Reviewers
Reviewers reached a consensus on the novelty of our motivation and the strength of our empirical results:
* **Novelty:** "The strongest aspect... is the empirical identification of the Q/K distribution shift... The query-centric clustering mechanism is a logical and novel solution." (**Reviewer 1**) / "Novel Query-Centric Indexing... a sound and valuable insight." (**Reviewer 4**)
* **Practicality:** "Tackles one of the most significant and practical challenges in LLM serving today." (**Reviewer 4**) / "Aligns well with realistic deployment patterns." (**Reviewer 2**)
* **Performance:** "Impressive combination of near-lossless accuracy at 95% sparsity." (**Reviewer 1**) / "Empirical robustness at extreme sparsity is convincing." (**Reviewer 2**)

## 3. Comprehensive Resolutions to Key Concerns
We have addressed **all** weaknesses raised by the reviewers. Below is a summary of our resolutions and new results.

### Concern 1: Robustness in Long-Generation Tasks (Chain-of-Thought)
*(Raised by Reviewers 1, 2, & 4)*
Reviewers asked whether our "offline" index remains effective when generating long sequences (e.g., CoT) where new queries might drift from the prefill distribution.

**Our Resolution:**
We conducted new experiments on **LongBench-v2** using **Chain-of-Thought (CoT)** decoding (generating up to **2048 tokens**).
* **Results:** As shown in the table 4 in paper (also included in our rebuttal), CSAttention remains **near-baseline** in accuracy, demonstrating strong transferability to generated queries.
* **Mechanism:** Our *streaming try-insert* strategy combined with a *recent-window passthrough* ensures the index adapts dynamically during generation.

### Concern 2: Baselines and Speedup Clarification
*(Raised by Reviewers 1 & 3)*
Reviewers requested a clearer definition of baselines and explicit speedup metrics against optimized kernels.

**Our Resolution:**
* **Baseline:** We clarified that our baseline is the standard **FlashAttention-2 (FA-2)** kernel.
* **Speedups:** We provided normalized speedup ratios for the **All-GPU regime** (Llama-70B, batch size 1). CSAttention achieves **1.59$\times$** (32k), **2.62$\times$** (64k), and **4.24$\times$** (128k) speedups over FA-2, confirming that our method's benefits grow significantly with context length.

### Concern 3: Deployment Scope and Prefill Overhead
*(Raised by Reviewers 3 & 4)*
Reviewers questioned the cost of building the index and the overhead of streaming updates during decoding.

**Our Resolution:**
* **Target Scenario:** We clarified that our method targets **"Offline Prefill / Online Decode"** workloads (e.g., RAG, Agents) where the prefill cost is amortized over many interactions.
* **Hiding Latency:** * In **CPU$\leftrightarrow$GPU mode**, the index update on CPU is completely **overlapped** with the PCIe transfer of KV data, hiding the latency.
    * In **All-GPU mode**, the update is a minor component ($\approx 10\%$ of step time) and runs in parallel CUDA streams.

### Concern 4: Hyperparameter Sensitivity
*(Raised by Reviewer 2)*
Reviewer 2 noted that the sensitivity of parameters like subspace count ($m$) and centroids ($C$) was underexplored.

**Our Resolution:**
We added **Appendix D** with a detailed sensitivity analysis.
* **Stability:** Results show performance is stable across a wide range (e.g., $m \in [6, 12]$).
* **Defaults:** We provided a "Recommended Defaults" table to ensure reproducibility and ease of use for future practitioners.

---

### Meta-Review · Area_Chair_bMkT · 2025-12-08

**Summary:**

This paper proposes CSAttention, a training-free, query-centric sparse attention method for long-context LLM inference. The method clusters queries during an offline prefill stage, precomputes centroid-key scores, and at decode time selects a small Top-K subset of keys for attention. It targets two regimes: a CPU-GPU, capacity-bound setting that reduces PCIe transfer, and an all-GPU, latency-bound setting that accelerates decoding via bounded lookups. Experiments on LongBench and LongBench-v2 report near-lossless accuracy at high sparsity and up to 4.24X speedup over FlashAttention-2 at long context lengths.

Across the four reviews, the paper was recognized as addressing an important practical problem and introducing a novel query-centric perspective. However, no reviewer clearly recommended acceptance (all scores are 4). Reviewers consistently raised concerns about the narrow applicability implied by strong offline-prefill assumptions, the additional memory and prefill overhead, and whether the claimed benefits justify the complexity and framing. The rebuttal added missing experiments and clarifications, but did not fully resolve these foundational concerns.

**Reviewer Concerns:**

Addressed:
* Added long-generation (CoT) evaluation on LongBench-v2, showing near-baseline accuracy under extended decoding.
* Clarified the dense baseline (FlashAttention-2) and provided normalized speedups at multiple context lengths.
* Added hyperparameter defaults, sensitivity analysis, and latency breakdowns.
* Clarified the intended offline-prefill / online-decode deployment scope.

Partially unresolved:
* The index still scales linearly with context length, so memory-capacity issues are not fundamentally resolved, making the "fixed-size" characterization misleading.
* Prefill overhead is high and restricts applicability to niche, amortized settings.
* The contribution appears primarily as a constant-factor engineering optimization with limited generality beyond the evaluated workloads.

**Reviewer Scores:**

* Reviewer WaZH (4): likely unchanged.
* Reviewer f8Pw (4): likely unchanged.
* Reviewer gZeQ (4): likely unchanged.
* Reviewer s5u9 (4): likely unchanged.

---

### Decision · Program_Chairs · 2026-01-26

Reject